# Comparative Analysis of Remote Sensing Storage Tank Detection Methods Based on Deep Learning

Lu Fan [1,2,*], Xiaoying Chen [1], Yong Wan [1] and Yongshou Dai [1]

1  College of Oceanography and Space Informatics, China University of Petroleum, Qingdao 266580, China; z22160095@s.upc.edu.cn (X.C.); wanyong@upc.edu.cn (Y.W.); daiys@upc.edu.cn (Y.D.)
2  Technical Test Centre of Sinopec, Shengli Oil Field, Dongying 257000, China
*  Correspondence: z22160104@s.upc.edu.cn or fanlu550.slyt@sinopec.com; Tel.: +86-178-5427-8192

**Abstract:** Since the Industrial Revolution, methane has become the second most important greenhouse gas component after $CO_2$ and the second most important culprit of global warming, leading to serious climate change problems such as droughts, fires, floods, and glacial melting. While most of the methane in the atmosphere comes from emissions from energy activities such as petroleum refining, storage tanks are an important source of methane emissions during the extraction and processing of crude oil and natural gas. Therefore, the use of high-resolution remote sensing image data for oil and gas production sites to achieve efficient and accurate statistics for storage tanks is important to promote the strategic goals of "carbon neutrality and carbon peaking". Compared with traditional statistical methods for studying oil storage tanks, deep learning-based target detection algorithms are more powerful for multi-scale targets and complex background conditions. In this paper, five deep learning detection algorithms, Faster RCNN, YOLOv5, YOLOv7, RetinaNet and SSD, were selected to conduct experiments on 3568 remote sensing images from five different datasets. The results show that the average accuracy of the Faster RCNN, YOLOv5, YOLOv7 and SSD algorithms is above 0.84, and the F1 scores of YOLOv5, YOLOv7 and SSD algorithms are above 0.80, among which the highest detection accuracy is shown by the SSD algorithm at 0.897 with a high F1 score, while the lowest average accuracy is shown by RetinaNet at only 0.639. The training results of the five algorithms were validated on three images containing differently sized oil storage tanks in complex backgrounds, and the validation results obtained were better, providing more accurate references for practical detection applications in remote sensing of oil storage tank targets in the future.

**Keywords:** remote sensing image; target detection; tank detection; deep learning

## 1. Introduction

Since the Industrial Revolution, widespread use of fossil fuels, deforestation, and other human activities have led to a continuous increase in greenhouse gases in the atmosphere, and the prolonged accumulation of large amounts of $CO_2$ and other greenhouse gases in the atmosphere has produced a greenhouse effect, resulting in a global temperature about 1.2 °C higher than that before industrialization [1]. Greenhouse gases are the main driver of climate change, bringing about serious climate problems such as droughts, fires, floods, and melting glaciers [2]. Climate change is a serious challenge shared by the whole world and poses a serious threat to the development and survival of human society, and addressing climate change has become a global consensus [3]. Since 2019, under the call of the United Nations Intergovernmental Panel on Climate Change (IPCC) Special Report on Global Warming of 1.5 °C, all major economies in the world have proposed net-zero emissions or carbon-neutral targets according to their own realities [4]. As the world's largest carbon emitter, China should actively fulfill and assume its obligations and responsibilities for carbon emissions reduction and climate change [3]. On 22 September 2020, at the general debate of the 75th UN General Assembly, President Xi Jinping announced to the world

that he would increase the country's autonomous contribution, adopt stronger policies and measures, and strive to peak carbon emissions by 2030 and work towards achieving carbon neutrality by 2060 [3,5].

Studies have shown that methane contributes up to 20% of the global temperature increase from anthropogenic greenhouse gas emissions since the Industrial Revolution and has become the most important greenhouse gas component after $CO_2$ [6]. Most methane is derived from emissions from energy activities, and petroleum refining and petrochemical processes are important sources of methane emissions [7–9]. Storage tanks, a type of equipment used to store crude oil and petroleum products, are an important component of crude oil and natural gas extraction processes and are one of the major sources of methane emissions [10–12]. Crude oil and natural gas stored in storage tanks usually contain large amounts of methane, which can escape into the atmosphere if the tanks are not effectively controlled and managed, resulting in greenhouse gas emissions and negative environmental impacts. Storage tank data can effectively reflect methane emissions from oil and gas production sites. Accordingly, the greater the number of storage tanks in an oil and gas production base, the more the methane emissions from that production base will increase. Therefore, using the spatial distribution of storage tanks, target identification and detection of storage tanks can help effectively count the greenhouse gas emissions of each oil and gas production base, thus effectively assisting the oil and gas industry to achieve energy savings and emissions reductions, which are of great significance to promote the strategic goal of "carbon neutrality and carbon peaking".

Traditional tank data statistics are mainly collected manually, which requires huge labor costs and capital. The sources of tank data include sales data from tank companies and statistics from oil and gas bases, which cannot simply and objectively provide indicators of tank spatial distribution. Therefore, there is an urgent need for convenient, efficient and accurate statistical methods to extract the spatial distribution of oil and gas base tanks in a timely and efficient manner.

In recent years, with the rapid development of remote sensing technology, the methods for acquisition of high-resolution remote sensing image data have become more and more diverse. High-resolution remote sensing images provide higher image quality and richer information detail, which provides great opportunities for the development of target detection in the remote sensing field [13]. Compared with traditional tank detection methods, efficient extraction of tank classification features from high-resolution remote sensing images can achieve automatic and efficient classification of target objects more conveniently and effectively. The current target detection methods for oil storage tanks are mainly divided into two types [14]: the first is based on the traditional machine learning target detection method for oil storage tank identification; the second is a deep learning-based detection method for oil storage tank identification. The traditional target detection method is divided into three parts: candidate area pre-selection, feature extraction, and classifier classification. Traditional machine learning detection algorithms' feature extraction is very dependent on manual selection, and there are problems such as poor detection, large computation, slow computing speed, and long training time. Therefore, traditional machine learning detection algorithms are no longer applicable to current-day large sample data sets [15]. Deep learning-based detection methods for oil storage tank identification introduce CNN [16] networks into the target recognition process, using data-driven feature extraction, and are able to obtain deep, dataset-specific feature representations based on learning a large number of samples. Deep learning-based detection algorithms are more efficient and accurate representations of datasets [17], and the extracted abstract features are more robust and have better generalizability.

Current algorithms for remote sensing image target detection based on deep learning can be divided into two main categories [18]: algorithms based on candidate regions [19–21] and those based on regression analysis [22,23], which are sometimes referred to as two-stage and single-stage algorithms, respectively [24]. The main difference between the two is that two-stage algorithms require first pre-selection of boxes and then object feature extrac-

tion, while single-stage algorithms will extract features directly in the network to predict object classification and location. The YOLOv5 [25] (you only look once), YOLOv7 [26] and SSD [27] (single shot multi box detector) algorithms are all end-to-end single-stage detection algorithms, and in general single-stage detection algorithms are fast in detection, but the detection accuracy often does not reach that of two-stage algorithms. However, RetinaNet [28] proposed a focal loss function to address the main reason for the difference in accuracy between single-stage and two-stage detection algorithms: Class Imbalance, which makes the detection accuracy of RetinaNet (single-stage target detection algorithm) exceed that of classical two-stage target detection networks. The main two-stage detection algorithms include RCNN [29] (regions with CNN features), Fast RCNN [30], and Faster RCNN [31]. The design of two target detection processes in two-stage detection algorithms improves the accuracy of the algorithm, but also increases the complexity and time cost of the model, which limits the computational efficiency. Single-stage target detection algorithms perform well in reducing time cost and improving computational efficiency, but are slightly inferior to two-stage algorithms in recognition efficiency for complex samples. Remote sensing images will have problems with scale diversity, large scale of small targets, dense target distribution, and high background complexity compared with conventional images. Therefore, general detection algorithms with excellent performance on conventional images at this stage do not necessarily have the same detection effect on remote sensing images as that on conventional data. This paper constructs a dataset containing different types of oil tanks in complex backgrounds, and based on the theoretical study and comparative analysis of various deep learning algorithms, combining the accuracy needs of the subject context (the higher the detection accuracy, the more statistically accurate the obtained methane information),we compare the effects of the above-mentioned deep learning algorithms in oil tank target detection through experiments in detail, and further validate the model training results on three images containing different sizes of oil tanks in complex backgrounds. This experiment provides a reference for practical detection applications in remotely sensed oil tank targets, and provides guidance and basis for further selection and improvement of related algorithms.

## 2. Algorithm Framework and Analysis

### 2.1. Faster RCNN

Ross B. Girshick proposed Faster RCNN in 2016 [31]. The Faster RCNN framework is divided into four main parts: feature extraction network, region candidate network (RPN), interest domain pooling, and classification and regression network, among which region suggestion network and interest domain pooling are the key elements of Faster RCNN. A structure diagram of a Faster RCNN network is shown in Figure 1. The backbone network of Faster RCNN is visual geometry group 16 (VGG16) [32], composed of thirteen $3 \times 3$ convolution layers, three fully connected layers, and several pooling layers. RPN (Region of Interest) is a fully convolutional neural network, which differs from ordinary convolutional neural networks by turning the fully connected layers in CNN into convolutional layers. The RPN network structure is shown in the red box in Figure 1. The RPN network is actually divided into two lines; the upper is used to obtain positive and negative sample classification via a SoftMax classification anchor box, and the lower is used to calculate the bounding box regression offset for anchors to obtain accurate anchor boxes. After inputting the test image, the whole image will be input into CNN for feature extraction; with RPN one first generates a number of Anchor boxes, crops and filters them, and then uses SoftMax to judge the anchors as belonging to foreground or background, that is, whether they are or are not objects, so this is a binary classification; meanwhile, another branch of the bounding box regression corrects the anchor box to form a more accurate proposal; the proposal window is mapped to the last layer of the convolutional feature map of the CNN; the ROI pooling layer makes each ROI generate a fixed size feature map; and finally, the classification probability and bounding box regression are jointly trained using SoftMax Loss (detecting classification probability) and Smooth L1 Loss (detecting

border regression). The Faster RCNN network directly uses RPN to generate candidate regions, which greatly improves the speed of generating regions to be detected. In addition, the Faster RCNN framework integrates the feature extraction network, the target candidate region selection network, and the classification regression task into the neural network, which realizes end-to-end detection in the two-stage detection algorithm and greatly improves the comprehensive performance of the Faster RCNN detection algorithm. The detection speed of the Faster RCNN detection algorithm is greatly improved and the detection time of the algorithm is reduced.

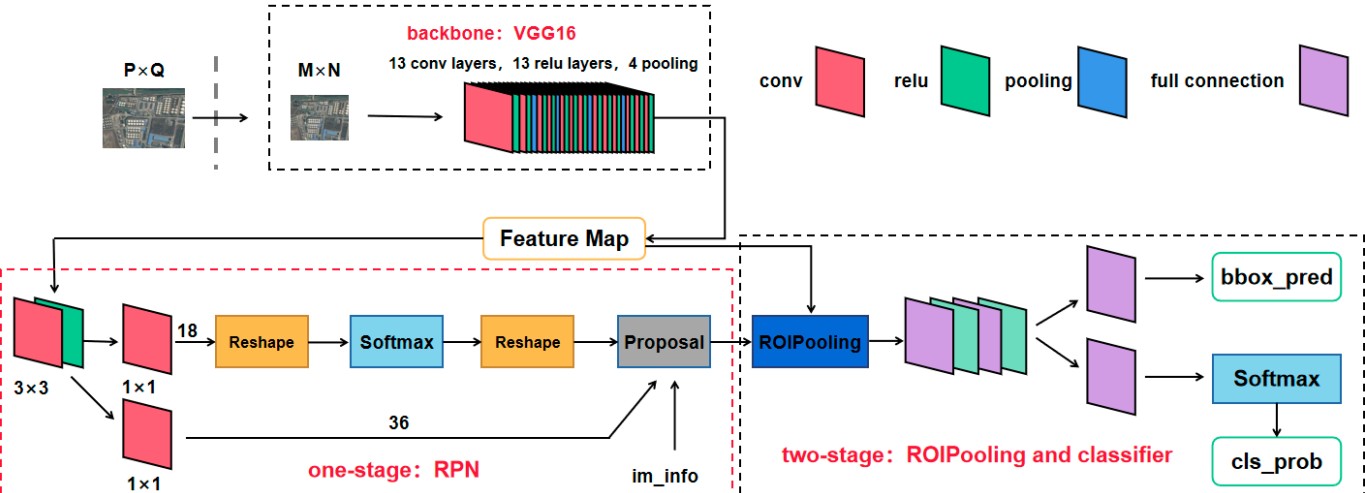

**Figure 1.** Faster RCNN network structure diagram.

## 2.2. YOLOv5

The YOLOv5 network, as a representative of single-stage detector, has obvious advantages in target detection speed. The YOLOv5 network consists of four main parts: input, reference network, neck network and head output layer. A structure diagram of the YOLOv5 network is shown in Figure 2. The input image size in the YOLOv5 network is $608 \times 608$, and this stage usually uses mosaic data enhancement, adaptive anchor frame calculation, and adaptive image scaling to pre-process the input image, i.e., the input image is scaled to the input size of the network and normalization and other operations are performed. The benchmark network is usually the network of some high-performance classifier species, and this module is mainly used to extract general feature representations. The YOLOv5 network uses the CSPDarknet53 structure with the focus structure as the benchmark network. The neck network is usually located in between the benchmark network and the head network, and it can be utilized to further improve the diversity and robustness of the features. YOLOv5 adds FPN (Feature Pyramid Network) [33] + PAN (Path Aggregation Network) [34] structures to the neck network part for multi-scale feature fusion of images; the FPN structure makes the bottom feature map contain stronger semantic information by up-sampling from the top down, and the PAN structure makes the top feature contain position information by down-sampling from the bottom up. The two features are finally fused to make feature maps of different sizes containing image semantic information and image feature information. The FPN + PAN structure is shown in Figure 3. The head output is used to complete the output of the target detection results. The number of branches on the output varies for different detection algorithms and usually contains a classification branch and a regression branch.

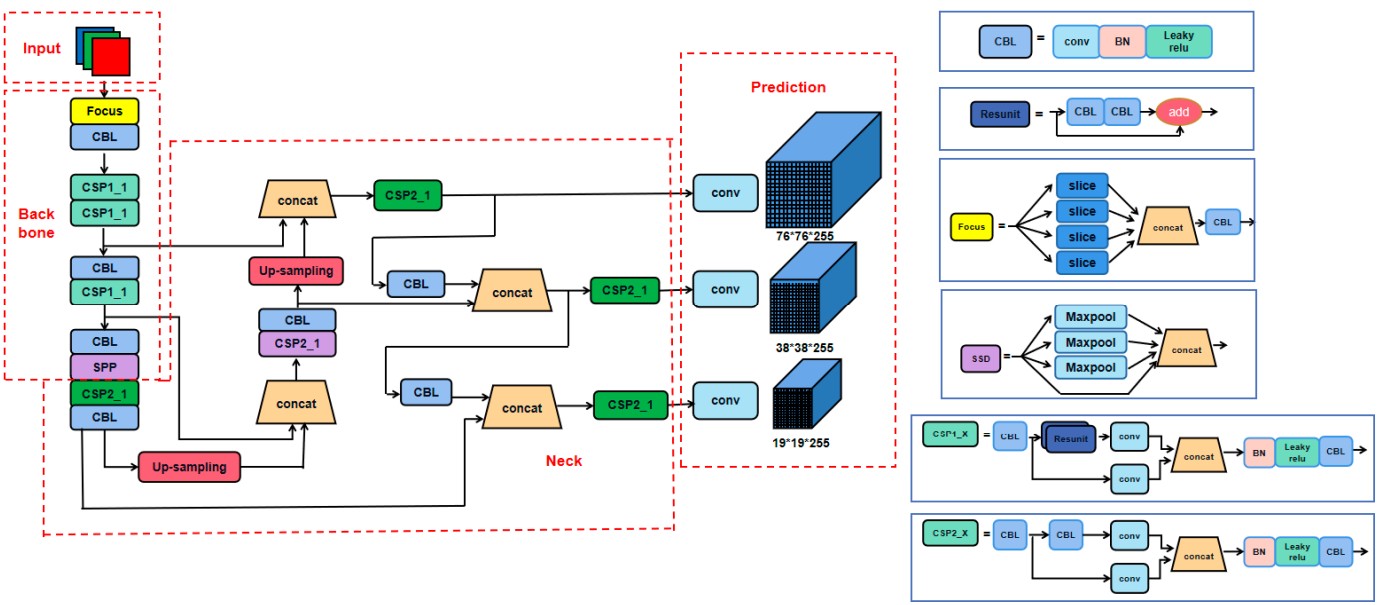

**Figure 2.** YOLOv5 network structure diagram.

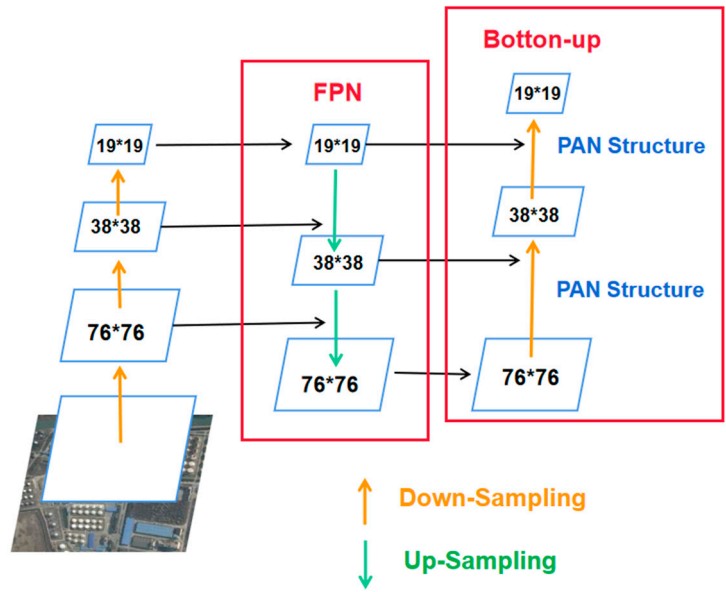

**Figure 3.** FPN + PAN structure diagram.

### 2.3. YOLOv7

The network framework of YOLOv7 basically follows the network structure of YOLOv5. YOLOv7 made some improvements based on YOLOv5, using more accurate cross-entropy, a more efficient way of marking assignments and a more efficient training method. First, YOLOv7 extends the high-efficiency process-enhanced focus internet, called Extended-ELAN (commonly known as E-ELAN) [26]. In a large-scale ELAN, the internet reaches an equilibrium state regardless of the gradient direction path length and total number of blocks. However, such equilibrium states may also be destroyed if endlessly cascading measurement blocks are used and the usage of the main parameters decreases. E-ELAN performs Expand, Shuffle, and Merge cardinality on cardinality to enable the network to learn more features by controlling the shortest and longest gradient paths without destroying the original gradient paths. Meanwhile, the YOLOv7 algorithm uses a concatenation-based model scaling method, the structure of which is shown in Figure 4. When performing model scaling on a cascade-based model, only the depth in the computational block needs

to be scaled, and the transition layer is scaled by the same amount of variation in the width factor, maintaining the characteristics of the model at the time of initial design. In addition, YOLOv7 proposes a training method for the auxiliary head, which greatly improves the detection accuracy of the algorithm by increasing the training cost.

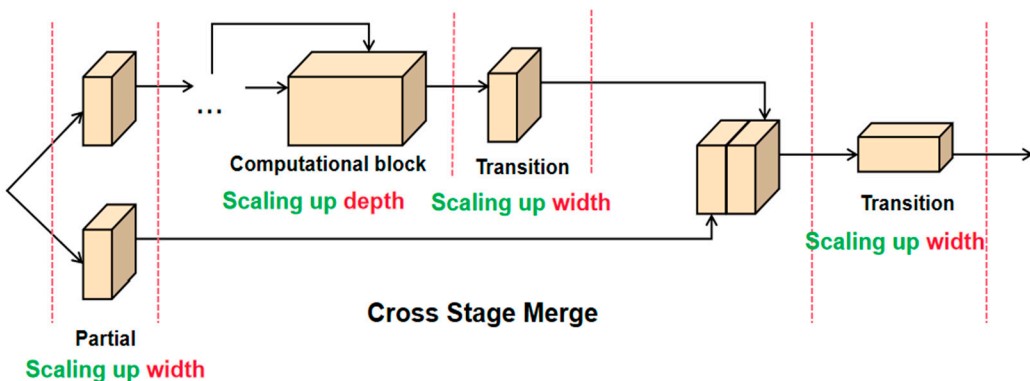

**Figure 4.** Structure diagram of the cascade-based model scaling method.

*2.4. RetinaNet*

Class imbalances are a major difficulty in training single-stage target detection models, the most serious of which is the imbalance between positive and negative samples. The RetinaNet detection algorithm proposes a loss function called Focal Loss for imbalance between foreground (positive) and background (negatives) categories in existing single-stage method target detection models [28]. Focal loss is a modified cross-entropy (CE) loss that multiplies the original CE loss [35] by an exponential that weakens the contribution of the easily detectable target to the model training, so that focal loss successfully solves the problem in which target detection loss is easily swayed by a large number of negative samples due to the extreme imbalance between positive and negative sample areas. The focal loss formula is shown below [28]:

$$FL(p_t) = -\alpha_t(1 - p_t)^\gamma \log(p_t)$$

$p_t$ is the classification probability of different categories, $\gamma$ is a value greater than 0, $\alpha_t$ is a fractional number between [0, 1], and is a fixed value that is not involved in training.

From the formula, it can be seen that the larger $p_t$ is, the smaller the $(1 - p_t)$ weight is, for both the foreground and background categories. When a sample category is clear, it contributes less to the overall loss, while if a sample category is not easily distinguishable, its contribution to the overall loss is relatively large. The resulting loss will eventually focus on inducing the model to try to distinguish those difficult target categories, thus effectively improving overall target detection accuracy. For adjusting the ratio of positive to negative samples, the foreground category is used when the corresponding background category is used; the optimal sum values are mutually influential, so the two need to be adjusted in combination when assessing accuracy.

The RetinaNet network consists of three main components: the baseline network (Backbone), the neck network and the head output layer. A structural diagram of the RetinaNet network is shown in Figure 5. The backbone of RetinaNet is the ResNet [36] network. The neck module of the RetinaNet network is the FPN network structure. The FPN module receives three feature maps, c3, c4, c5, and outputs five feature maps P2–P7, all with 256 channels and stride (8, 16, 32, 64, 128), where large stride (small feature map) is used to detect large objects and small stride (large feature map) is used to detect small objects. The head module consists of two branches, classification and location detection, each consisting of four convolutional layers, and the Head module weights are shared among the five output feature maps.

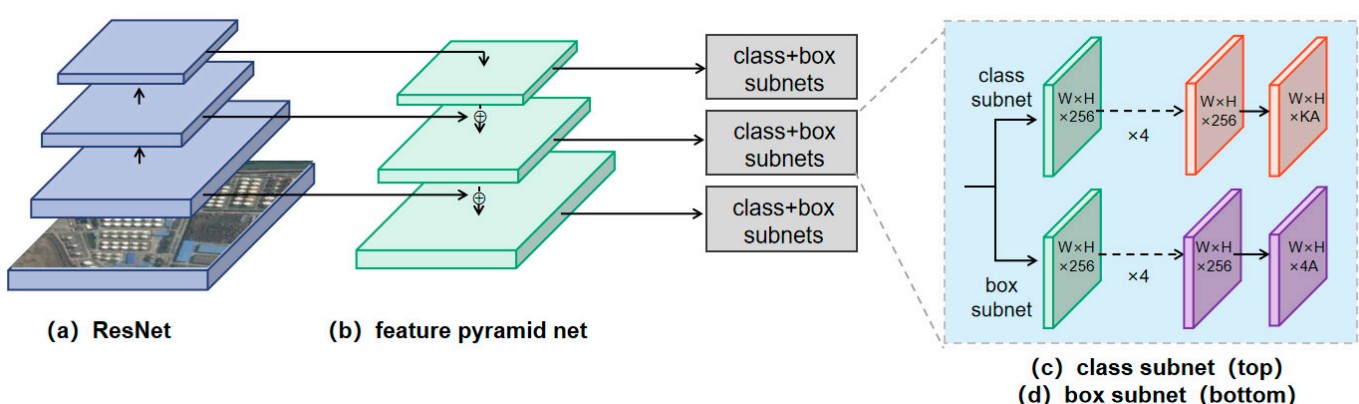

**Figure 5.** Structure diagram of the RetinaNet network.

The RetinaNet network uses the residual network ResNet to solve the problem of gradient explosion or disappearance during training due to the network being too deep. In addition, RetinaNet uses a top-down FPN structure to perform multi-scale fusion of the extracted feature maps, which enables it to maintain sufficient detection accuracy even for smaller objects.

### 2.5. SSD

The SSD algorithm [27] uses the same single convolutional neural network as YOLO to perform detection, but the YOLO algorithm suffers from problems of small target detection and inaccurate localization, which are overcome to some extent by the SSD algorithm.

SSD uses VGG16 as the base model, and then adds a new convolutional layer to VGG16 to obtain more feature maps for detection. The network structure of SSD is shown in Figure 6. Images of size $300 \times 300 \times 3$ are input to the VGG16 network to obtain feature maps of different sizes; the feature maps of the Conv4_3, Conv7, Conv8_2, Conv9_2, Conv10_2, Conv11_2 layers are extracted, and six default boxes of different scales are constructed at each point above these feature map layers. Then, the boxes are detected and classified separately to generate multiple initial qualified default boxes. Finally, the default boxes obtained from different feature maps are combined and suppressed using the NMS (non-maximum suppression) method [37] to eliminate some of the overlapping or incorrect default boxes and generate the final set of default boxes (i.e., the detection result).

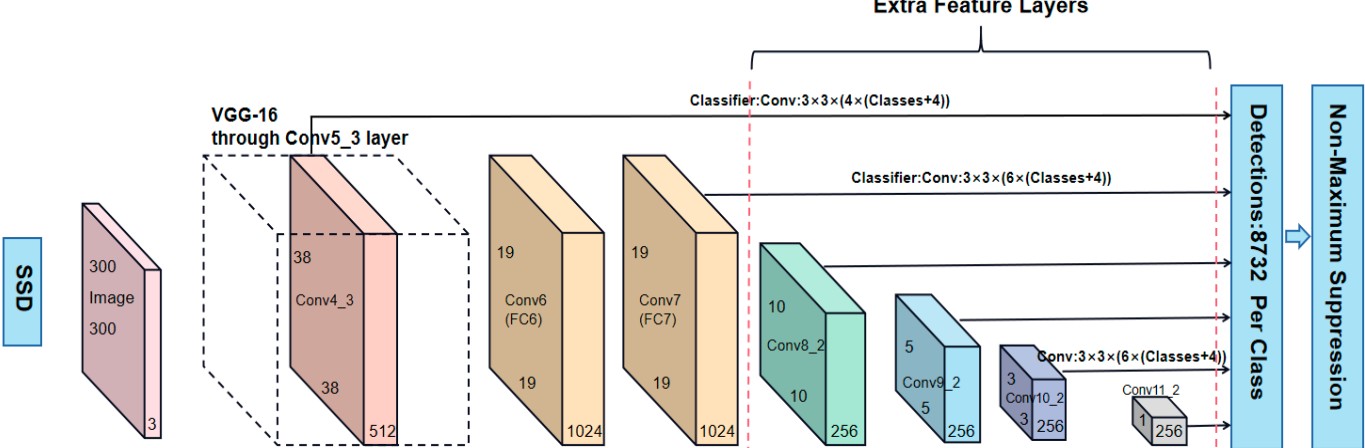

**Figure 6.** SSD network structure diagram.

SSD uses multiple feature layers with sizes of $38 \times 38$, $19 \times 19$, $10 \times 10$, $5 \times 5$, $3 \times 3$, and $1 \times 1$. The large size feature map uses shallow information to predict small targets;

the small size feature map uses deep information to predict large targets. This multi-scale detection approach can make detection more adequate (SSD performs dense detection) and better detect small targets.

The SSD algorithm uses a regression analysis-based algorithm to directly regress the class and location of the object while also utilizing the idea of a candidate region-based algorithm to generate candidate target frames. The SSD algorithm balances the advantages and disadvantages of the YOLO model and the Faster RCNN model when taking into account the detection accuracy and training speed of the model. The main idea behind SSD network (full convolutional network) design is featuring extraction in layers and edge regression and classification in turn. Because different levels of feature maps can represent different levels of semantic information, low-level feature maps can represent low-level semantic information (containing more details), which can improve the quality of semantic segmentation and are suitable for learning small-scale targets. High-level feature maps can represent high-level semantic information with smooth segmentation results, and are suitable for in-depth learning of large-scale targets.

## 3. Experiment and Analysis

### 3.1. Experimental Environment and Data

This experiment was conducted on a 64-bit Win10 system, with NVIDIA RTX A4000 GPU, 16 GB video memory, Python version 3.7, and PyTorch and Keras as the deep learning framework.

Due to the fact that very few existing remote sensing datasets contain oil tank targets alone, most remote sensing datasets contain multiple targets (e.g., aircraft, playgrounds, oil tanks, overpasses, etc.), and the number of remote sensing images of oil tanks contained in any single multi-objective remote sensing dataset is not enough to support training a deep learning target algorithm. Therefore, several remote sensing datasets were selected for this experiment, and the target images and label files of oil tanks required for the experiment were selected from these datasets according to the target information in their respective label files.

In this experiment, tank images from five different datasets were used for training, including the DIOR dataset [38], the NWPUU_RESISC45 dataset [39], the NWPU VHR-10 [40] dataset, the TGRS-HRRSD dataset [41], and a self-built dataset. The DIOR dataset is a large-scale benchmark dataset for optical remote sensing image target detection. The dataset contains 23,463 images and 192,472 instances, covering 20 object classes such as aircraft, airport, baseball field, basketball court, storage tank, etc. The NWPU-RESISC45 Dataset is a usable benchmark for remote sensing image scene classification created by Northwestern Polytechnic University, which contains a total of 31,500 images with pixel size of 256 × 256, covering 45 scene categories. The NWPU VHR-10 dataset is an open-source remote sensing image target detection dataset released by Northwestern Polytechnic University in 2014, which contains 10 geospatial object classes such as aircraft, baseball field, basketball court, storage tank, tennis court, etc. The TGRS-HRRSD dataset is a dataset released by Xi'an Institute of Optical Precision Machinery, Chinese Academy of Sciences in 2017 for the study of high-resolution remote sensing image target detection, which contains 13 types of remote sensing ground object targets. The self-built dataset contains satellite images of industrial areas with oil tanks around the world taken from Google Earth. This experiment used Labelimg software to label the images of the tank targets in the self-built remote sensing image dataset. Figure 7 shows Labelimg labeled images. Basic information about the sample dataset is shown in Table 1, and the oil tank dataset constructed in this paper covers different types of oil tanks in various complex backgrounds. Finally, this experiment integrated data from the above five datasets in a randomized arrangement, which ensured randomness and accuracy in this experiment and also increased the robustness of the algorithm accordingly.

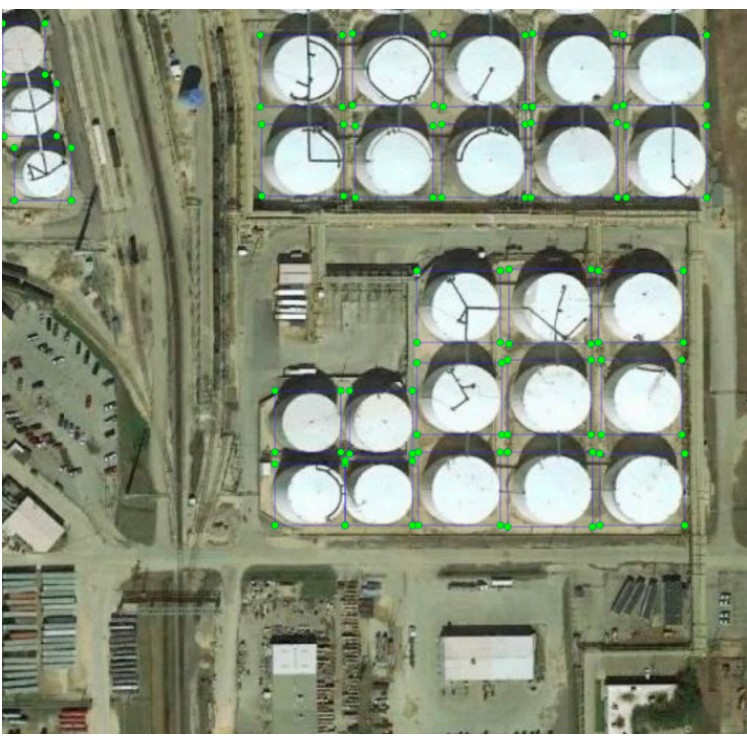

**Figure 7.** Labelimg tagged images.

**Table 1.** Basic information about the datasets.

| Dataset | Image Resolution | Image Size Range/Pixels | Number of Images Containing Tanks | Number of Targets (Tanks) |
|---|---|---|---|---|
| DIOR | 0.5~30 m | 800 × 800 | 1244 sheets | 20,361 |
| NWPUU_RESISC45 | 0.5~30 m | 256 × 256 | 688 sheets | 12,405 |
| NWPU VHR-10 | 0.5~2 m | (500~1100) × (500~1100) | 165 sheets | 1698 |
| TGRS-HRRSD | 0.15~1.2 m | (152~10,569) × (152~10,569) | 897 sheets | 4406 |
| self-built dataset | 0.5~3 m | 512 × 512 | 574 sheets | 7205 |
| Total | - | - | 3568 sheets | 46,075 |

*3.2. Network Performance Evaluation*

In the model evaluation task of this study, the precision, recall, F1 score, and mAP of five target detection network training models were calculated to clearly and accurately compare the recognition performance of the five detection models in numerical terms.

mAP, mean average precision, generally refers to the average of the APs of all categories within all pictures. mAP is considered by most studies to be the most important performance evaluation metric in target detection model evaluation. mAP is obtained based on average precision (AP); assuming that C categories need to be detected, the specific formula for mAP is shown below:

$$mAP = \frac{\sum\limits_{i=1}^{C} AP_i}{C}$$

AP refers to the area under the PR curve (precision–recall curve) for a specific category within all images. The mean accuracy (mAP) is the average of the AP of multiple entity categories. The value of mAP in the experiment is between 0 and 1. A high mAP score is associated with superior model recognition performance. This metric is crucial for evaluating the performance of target detection algorithms. As this experiment selected

only one class of target entities, mAP and AP have the same value in calculating the evaluation metric.

According to the conceptual definition of the target entity binary classification problem, the real category of the target entity and the category predicted by the classification algorithm can be set as the evaluation criteria, and the specific classification criteria can be referred to the confusion matrix in Figure 8.

| Confusion Matrix | | Predicted value | |
|---|---|---|---|
| | | Positive | Negative |
| **Real Value** | Positive | TP | FN |
| | Negative | FP | TN |

**Figure 8.** Confusion matrix.

The four parameters in the table are described as follows:

True Positive (TP): predicted positive case, actual positive case; the algorithm predicts correctly (True)

False Positive (FP): predicted positive case, actual negative case; that is, the algorithm predicts wrong (False)

True Negative (TN): the prediction is negative, the actual case is negative; that is, the algorithm predicts correctly (True)

False Negative (FN): the prediction is negative, the actual case is positive; that is, the algorithm predicts wrong (False)

The judgment conditions of positive and negative cases need to be determined by the IOU values, where IOU (Intersection over Union) refers to the ratio of the intersection area of the ground truth bbox and the predict bbox to the area of their concurrent sets. The larger the IOU value, the better the performance of the model algorithm for the prediction detection frame. Usually, regions with IOU $\geq$ 0.5 are set as positive examples (target) and regions with IOU $\leq$ 0.5 are set as negative examples (background) in the target detection task. Figuratively speaking, the IOU can be explained with Figure 9:

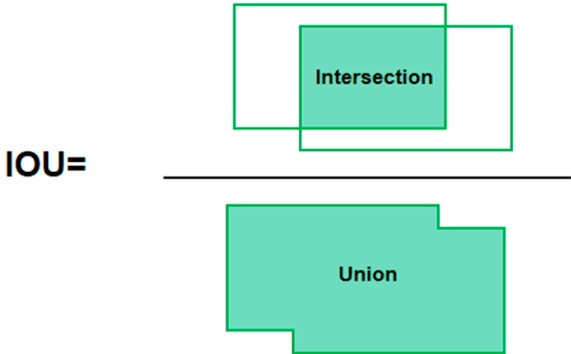

**Figure 9.** IOU schematic.

The precision, recall and F1 score formulas are defined as follows:

$$\text{Precision} = \frac{\text{TP}}{\text{TP} + \text{FP}} = \frac{\text{TP}}{\text{Samples with a positive prediction}}$$

$$\text{Recall} = \frac{\text{TP}}{\text{TP} + \text{FN}} = \frac{\text{TP}}{\text{true for positive samples}}$$

$$\text{F1 Score} = \frac{2 \times \text{precision} \times \text{recall}}{\text{precision} + \text{recall}}$$

The F1 score is the summed average of precision and recall and is used to measure precision and recall accuracy in a binary classification model as well as in other classification models. F1 score is defined as the summed average of precision and recall, which takes into account the accuracy of model recognition as reflected by the precision and recall values, using the recall when the two metrics conflict.

In summary, precision measures whether the model is misidentified and recall measures whether the model misses the target entity. MAP and F1 scores take into account both precision and recall and reflect the combined performance of the trained model.

### 3.3. Experimental Results and Analysis

In this experiment, 3568 remote sensing images were used, and the dataset was divided into a training set and a test set at a ratio of 8:2, where the training set contained 2854 data and the test set contained 714 data. The Faster RCNN, YOLOv5, YOLOv7, RetinaNet and SSD algorithms were tested using the dataset. In order to accurately predict the detection effect of the algorithm, the number of oil tank detections was counted with a confidence level of 0.5. Multiple sets of precision and recall were obtained in different confidence intervals, and the average precision mAP values and F1 scores were calculated for accuracy evaluation [42–44].

Statistics of the detection results are shown in Table 2. The average accuracy and F1 scores of SSD reached 0.897 and 0.870, which indicate accurate detection of most of the tank targets, show high detection accuracy and robustness with less false detection and leakage detection, and were the best results under the experimental conditions. The detection accuracy of the YOLOv7 algorithm was second only to that of SSD, reaching 0.870 with a F1 score of 0.826, except for very small targets in the image, which could not be detected accurately. Other conventional and large-scale targets were detected accurately, with fewer false detections and missed detections. The average accuracies of YOLOv5 and Faster RCNN were close, both above 0.84; the F1 score of Faster RCNN was 0.778 and the F1 score of YOLOv5 was 0.850, among which Faster RCNN could detect more tanks, but there were some false detections. The average accuracy of RetinaNet was only 0.639; the detection performance was poor, and the total number of detections was low.

**Table 2.** Statistical table of testing results for each algorithm.

| Algorithm Name | Backbone Network | Deep Learning Framework | Precision | Recall | F1 | $\text{map}_{0.5}$ | $\text{map}_{0.5\sim0.9}$ |
|---|---|---|---|---|---|---|---|
| Faster RCNN | VGG16 | Pytorch | 0.826 | 0.735 | 0.778 | 0.842 | 0.629 |
| YOLOv5 | CSP-Darknet53 | Pytorch | 0.800 | 0.907 | 0.850 | 0.847 | 0.613 |
| YOLOv7 | Resnet-101 | Pytorch | 0.839 | 0.806 | 0.826 | 0.870 | 0.632 |
| RetinaNet | Resnet-50 | Kreas | 0.804 | 0.762 | 0.782 | 0.639 | 0.406 |
| SSD | VGG16 | Pytorch | 0.865 | 0.876 | 0.870 | 0.897 | 0.641 |

The loss function is used to estimate the degree of inconsistency between the predicted and true values of a model, and it is a non-negative real-valued function; usually, the smaller the loss function, the better the robustness of the model [39]. Figure 10 shows the loss value curves of the five target detection algorithms during training.

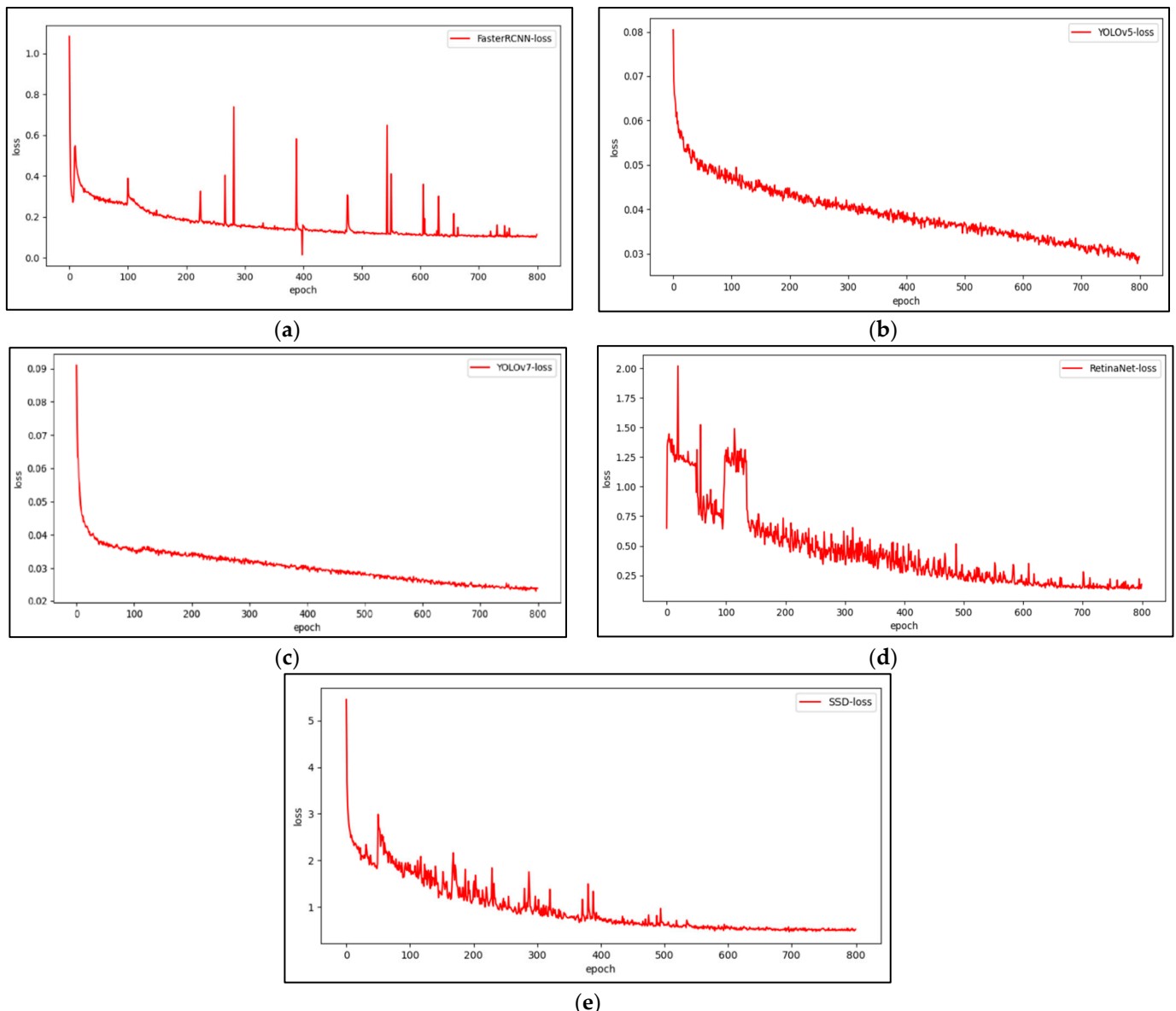

**Figure 10.** Variation in loss values during the training of the five target detection algorithms: (**a**) Faster RCNN loss value variation graph; (**b**) YOLOv5 loss value variation graph; (**c**) YOLOv7 loss value variation graph; (**d**) RetinaNet loss value variation graph; (**e**) SSD loss value variation graph.

The loss values are crucial for assessing the training quality of the algorithms, and models that present flat, low-level loss values have higher accuracy. As can be seen in Figure 10, the loss values of the training sets of the five target detection algorithms as a whole continued to decrease and stabilize as the number of training rounds increased. The loss values of the Faster RCNN and RetinaNet algorithms fluctuated more during the training process, but tended to be flat and stable during the final stage of the training cycle, so the model files at the final stage were selected. The loss functions of the YOLOv5, YOLOv7, and SSD algorithms were flat and stable overall, and the model files of the last 100 rounds were selected as the analysis model files for the next stage of experimental validation in order to make the results of the validation experiments more accurate.

In order to verify the accuracy of the training results of different algorithms, this study used the training weight files selected above for experimental validation. Three images containing multiple sizes of oil storage tanks against a complex background were selected for testing, where image 1 contained some small tank targets, image 2 contained some tanks

with similar color to the background, and image 3 contained many small targets similar to oil tanks. The image test results are shown in Figure 11.

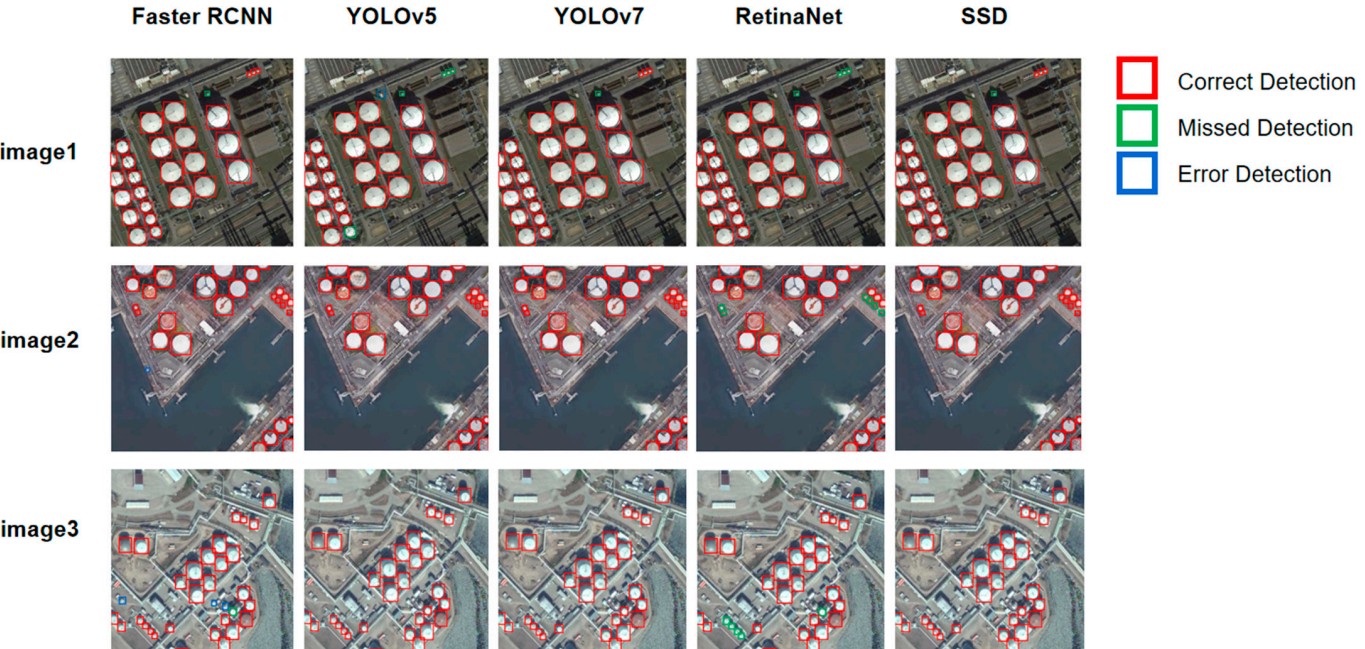

**Figure 11.** Schematic diagram of the detection effect of the different algorithms against complex backgrounds.

## 4. Discussion

This section analyzes the training results (recall, precision metrics, F1 score and mAP) and validation result image pairs based on the Faster RCNN, YOLOv5, YOLOv7, RetinaNet and SSD algorithms from the previous section. Precision indicates the proportion of samples predicted to be positive among those with true category as positive; recall indicates the proportion of samples successfully predicted by the model among those with true positivity. The F1 score is the average of the summation of precision and recall.

For the Faster RCNN algorithm, although the accuracy score is high, recall is low, indicating that the trained model has excellent accuracy in predicting the entities selected as target entities by the recognition frame, but fails to recognize all the necessary entities. While the SSD algorithm, YOLOv5 algorithm and YOLOv7 algorithm received good scores for accuracy and recall, the YOLOv5 model also still has shortcomings in predicting correct entities in this study compared to the SSD and YOLOv7 algorithms. The precision, recall, F1 score and mAP of RetinaNet algorithm are 0.804, 0.762, 0.782, 0.639, respectively; from the experimental data, the RetinaNet algorithm has poor overall performance in this experiment.

RetinaNet introduces focal loss for the imbalance problem between foreground (positive) and background (negative) categories in existing single-stage method target detection models, which reduces the impact of sample imbalance (positive and negative sample imbalance and difficult sample imbalance) in single-stage algorithms. However, focal loss is susceptible to noise interference, so the RetinaNet algorithm requires very high accuracy of image labeling, and once there are mislabeled samples, they will be identified as difficult samples by focal loss; therefore, interference samples contribute significantly to the loss and will have a great impact on the learning and training effect. The data used in this experiment are from five different datasets, and there are some differences in the labeling criteria for each dataset, which is the reason for the poor performance of the RetinaNet algorithm in this experiment. SSD and YOLO have obvious speed advantages as single-stage detectors, but in this experiment, the detection performance of SSD is significantly

better than that of YOLOv5 and YOLOv7. This is because the SSD algorithm uses the anchor generation mechanism from the Faster-RCNN algorithm to ensure the detection accuracy of the algorithm, and the regression idea from the YOLO algorithm to improve the training speed and detection speed. However, the SSD algorithm cannot accurately identify some small targets. This is because in the SSD algorithm, a prior box is generated for each pixel of the feature map, and SSD trains the feature map through the prior box. In the SSD training process, the IOU between the prior box and the ground truth must reach 0.5 before it is put into the network for training. A large target is likely to have a much larger ROI value and therefore contain more prior boxes, which can be trained adequately. By contrast, small targets will have much fewer prior boxes for training and will not be trained sufficiently, which will cause a problem of poor recognition of small targets by the SSD algorithm. From the experimental results, the detection accuracy of the YOLOv7 algorithm is 0.023 higher than that of YOLOv5. This is due to the fact that YOLOv7 made a series of improvements to the basis of the YOLOv5 algorithm; YOLOv7 adopts a deeper network structure, more accurate cross entropy, more efficient logo assignment method and more efficient training method. In addition, YOLOv7 also introduces some new technical means, which further improve accuracy. The faster RCNN algorithm, as a two-stage detection algorithm, has good detection ability, but due to its convolutional extraction network to extract low resolution data from the single-layer feature map, and Faster RCNN using the original ROI pooling twice, rounding will bring accuracy loss. As a result, the Faster RCNN algorithm is not a good solution to the problem of multi-scale and small target detection, and is not ideal for detecting smaller objects in the above test images.

## 5. Conclusions

Since the Industrial Revolution, methane has become the second most important greenhouse gas component after $CO_2$. The long accumulation of large amounts of $CO_2$, methane and other greenhouse gases in the atmosphere produces a greenhouse effect, leading to global warming and bringing serious climate change problems such as droughts, fires, floods, and glacial melting. Most of the methane in the atmosphere comes from emissions from energy activities such as petroleum refining, and storage tanks are an important source of methane emissions during crude oil and natural gas extraction and processing. Traditional tank statistics are complex and expensive to obtain and do not provide simple and objective indicators of tank spatial distribution. With the rapid development of remote sensing technology, the use of high-resolution remote sensing image data to achieve efficient and accurate statistics for oil and gas production sites is important to promote the strategic goal of "carbon neutrality and carbon peaking".

In this paper, we constructed a dataset with different types of oil storage tanks in complex backgrounds, and conducted comparative experiments in oil storage tank detection on the constructed dataset using five algorithms, Faster RCNN, YOLOv5, YOLOv7, RetinaNet, and SSD, and came to the following conclusions.

(1) SSD has the best detection effect compared with other algorithms and is less affected by tank size and background complexity, with good robustness and higher detection accuracy; F1 scores are also higher. The average accuracy of three algorithms, Faster RCNN, YOLOv5, and YOLOv7, was above 0.84, and all achieved good F1 scores. Among them, YOLOv7, as a single-stage target detection algorithm, achieved good performance in detection accuracy while taking advantage of its own detection speed. The average accuracy of YOLOv5 and Faster RCNN was close, but the F1 score of YOLOv5 was 0.072 higher than that of Faster RCNN, where Faster RCNN had excellent accuracy in predicting target entities, but there were some false detections. The average accuracy of RetinaNet was only 0.639, so the detection performance was poor and the total number of detections was low.

(2) In the process of optical remote sensing detection of oil storage tanks, the quality of the image, the complexity of the background and the size of the tank target in the image have some influence on the detection results. Through experiments, it could be found

that the above algorithms are not effective in detecting small and medium-sized targets in images, and solving the problem of detecting multi-scale and small targets in remote sensing images is the next focus of research into optical remote sensing oil storage tank detection based on deep learning.

This study applies deep learning target detection techniques based on high-resolution remote sensing image data to the process of identifying and counting storage tanks in natural gas and oil field bases, focusing on the performance of five target detection algorithms with different principles in the process of identifying storage tanks.

**Author Contributions:** Data curation, L.F. and X.C.; investigation, Y.W.; resources, Y.W., and Y.D.; writing—original draft preparation, X.C.; writing—review and editing, L.F. All authors have read and agreed to the published version of the manuscript.

**Funding:** This study is supported by 1. Research project on monitoring, tracing and accounting system of methane and VOCs in oilfields funded by Technical Testing Center of Shengli Oilfield Branch, China Petroleum & Chemical Corporation, China, grant number: 322121. 2. Time-varying Seismic Subwave Extraction and Inverse Folding Methods under Complex Fading and Noise Interference, grant numbers: 41974144. 3. Research on Deep Learning-based Seismic Subwave Extraction Method in Space-Time Domain before Deep Earth Stacking, grant numbers: 42274159.

**Data Availability Statement:** Publicly available datasets were analyzed in this study. These data can be found here: [https://aistudio.baidu.com/aistudio/datasetdetail/53045], [https://aistudio.baidu.com/aistudio/datasetdetail/51873], [https://aistudio.baidu.com/aistudio/datasetdetail/52812] and [https://github.com/CrazyStoneonRoad/TGRS-HRRSD-Dataset] (all accessed on 7 May 2023).

**Acknowledgments:** Thanks to Northwestern Polytechnical University for remote sensing image data. Thanks to Google Earth and Baidu Map and kaggle website for remote sensing image data.

**Conflicts of Interest:** The authors declare no conflict of interest.

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
