# Peer review of "Comparative Analysis of Remote Sensing Storage Tank Detection Methods Based on Deep Learning"

_remotesensing, doi:10.3390/rs15092460_

Round 1
Reviewer 1 Report
The article presents an important topic, namely the detection of oil storage tanks using deep learning methodologies. While the subject is significant, plenty of literature already exists on it. Therefore, clarifying the objectives and reasoning behind the work would be beneficial.
Regarding the manuscript's content, please find below some suggestions that could improve the overall quality of the composition.
- Some sentences appear too vague and need a revision, e.g. "...machine learning detection algorithm has achieved certain results...".
- The title should be more specific and closely aligned with the nature of the article, which is a comparison of different algorithms on a specific issue. This will make it easier for readers to understand the focus of the research.
- The introduction should provide a broader and more detailed explanation of the "inestimable role in guaranteeing national defense security" and "carbon neutrality and carbon peaking" mentioned in the manuscript. This will help readers grasp the significance of these concepts in the research context.
- To facilitate comparison, the deep learning algorithms and frameworks should be introduced and explained using a consistent structure. This will make it easier for readers to understand the similarities and differences between the algorithms.
- Datasets are a crucial component of the research; therefore require a minimum introduction and description, including references. Additionally, a specific section should be dedicated to the processes needed to merge such datasets. This will clarify the reasons for mentioning the "construction" of a dataset.
- The conclusions should be updated and expanded to reflect the proposed changes in the manuscript. This will provide a clear and concise summary of the research.
- Finally, the list of references should be expanded and included throughout the manuscript, particularly in the introduction and chapter two. This will provide readers with the necessary background information and support for the research.
In summary, while the manuscript addresses an important topic, improvements can be made to enhance the overall quality of the composition. By clarifying the objectives and reasoning behind the research, providing more detailed explanations of critical aspects, and including necessary references, the manuscript will be more accessible to a broader audience.
The quality of the English is sufficient, but a minor revision is recommended. In particular, please check the pertinence of some verbs throughout the manuscript.
Reviewer 2 Report
Research on remote sensing oil storage tank detection method based on deep learning is presented. It is interesting but not complete enough for publication. Specific comments:
(1) Some spelling mistakes and writing style can be further improved, these small mistakes need to be corrected before the paper is published.
(2) Some details of the algorithm were not elaborated in detail, making it difficult for readers to grasp.
(3) It only focuses on five deep learning detection algorithms and does not compare them with other types of algorithms.
(4) It only validates the training results on three images containing different sizes of oil storage tanks in complex backgrounds. Therefore, it may not be generalizable to other types of images or backgrounds.
(5) There were few comparative experiments, and the content of the article was not rich enough.
(6) The experimental analysis was too simple and more comparative indicators could be added.
(7) In the introduction section, the author omitted some important references about deep learning-based methods in the field of remote sensing, such as
https://www.doi.org/10.1016/j.isprsjprs.2020.12.015
https://www.doi.org/10.1109/TGRS.2023.3248040
Minor editing of English language required.
Reviewer 3 Report
This paper presents presents the comparison of five algorithms used for deep-learning-based remote sensing of large oil tanks. The paper is well structurized, has good descriptions and adequate conclusions.
Few minor comments/issues have been found:
Lines 34-38, 39-41, 87, 164-167, 172 - some references would be beneficial (even if they are repeated in the next paragraphs);
Lines 39-41 - only one type is mentioned in this sentence; what is the second type? (The types should be introduced one after another, and then described in greater detail).
Figure 6 - the box colour description could be fitted inthe figure's description to enhance its readability.
Round 2
Reviewer 1 Report
I am pleased to have the opportunity to review your updated manuscript, which has undergone significant improvements since the last round of revisions. I appreciate the effort you have put into addressing the previous comments and expanding the scope of the paper. The article now presents a more structured literature review, highlighting the key findings and gaps in knowledge in this area. The new sections solve almost all the critical points identified in the first round of revision. The manuscript can now be considered well-written and well-organised, contributing to the literature on the topic. Thank you for the opportunity to review your manuscript, and I look forward to seeing it published.
Reviewer 2 Report
I have no other questions.